

# 1 Winter hydrometeorological extreme events modulated by
# 2 large scale atmospheric circulation in southern Ontario

Olivier Champagne[1]*, Martin Leduc[2], Paulin Coulibaly[1,3], M. Altaf Arain[1]
1 School of Geography and Earth Sciences, McMaster University, Hamilton, Ontario, Canada
2 Ouranos and Centre ESCER, Université du Québec á Montréal, Montréal, Québec, Canada
3 Department of Civil Engineering, McMaster University, Hamilton, Ontario, Canada
Correspondence to: Olivier Champagne (champago@mcmaster.ca)
**Abstract.** Extreme events are widely studied across the world because of their major implications for many
aspects of society and especially floods. These events are generally studied in term of precipitation or temperature
extreme indices that are often not adapted for regions affected by floods caused by snowmelt. Rain on Snow index
has been widely used but it neglects rain only events which are expected to be more frequent in the future. In this
study we identified a new winter compound index and assessed how large-scale atmospheric circulation controls
the past and future evolution of these events in the Great Lakes region. The future evolution of this index was
projected using temperature and precipitation from the Canadian Regional Climate Model Large Ensemble
(CRCM5-LE). These climate data were used as input in PRMS hydrological model to simulate the future evolution
of high flows in three watersheds in Southern Ontario. We also used five recurrent large-scale atmospheric
circulation patterns in northeastern North America and identified how they control the past and future variability
of the newly created index and high flows. The results show that daily precipitation higher than 10mm and
temperature higher than 5°C were a necessary historical condition to produce high flows in these three watersheds.
In the historical period, the occurrences of these heavy rain and warm events as well as high flows were associated
to two main patterns characterized by high Z500 anomalies centred on eastern Great Lakes (HP) and the Atlantic
Ocean (South). These hydrometeorological extreme events will be more frequent in the near future and will still
be associated to the same atmospheric patterns. The future evolution of the index will be modulated by the internal
variability of the climate system as higher Z500 in the east coast will amplify the increase in the number of events,
especially the warm events. The relationship between the extreme weather index and high flows will be modified
in the future as the snowpack reduces and rain becomes the main component of high flows generation. This study
shows the values of CRCM5-LE dataset to simulate hydrometeorological extreme events in Eastern Canada and
to better understand the uncertainties associated to internal variability of climate.



## 1 Introduction

According to the actual pathway of greenhouse gases emissions, temperature will continue to rise in the future with serious implications for society (Hoegh-Guldberg et al., 2018). The amount of precipitation, especially for extreme events, is also projected to increase globally (Kharin et al., 2013), due to the acceleration of the hydrological cycle (Trenberth, 1999). Because extreme precipitation has a great societal impact across the world, internationally coordinated climate indices, built from precipitation and temperature data, are widely used to assess the evolution of different weather extremes (Zhang et al., 2011). Some of these indices such as monthly or annual maximum of precipitation can be used to improve flood management. However, in catchments that receive snowfall, a large number of floods may occur due to a combination of temperature and precipitation extreme events such as Rain on Snow (ROS) (Merz and Blöschl, 2003). The impact of ROS on floods generation have been widely studied in different regions of the world, including Central Europe (Freudiger et al., 2014), the Alps (Würzer et al., 2016), the Rocky mountains (Musselman et al., 2018) or the New York State (Pradhanang et al., 2013). The projections of these events can be a challenge because they depend on the ability of the climate model to project the precipitation extremes and the aerial extent of snowmelt (McCabe et al., 2007). The climate models improvements allowed recent studies to project the future evolution of ROS (Il Jeong and Sushama, 2018; Musselman et al., 2018; Surfleet and Tullos, 2013). However strong uncertainties in the projections of such events remains, especially due to the internal variability of climate (Lafaysse et al., 2014). These uncertainties, even with the perfect climate model, will never be eradicated due to the inherently chaotic characteristic of the atmosphere (Lorenz, 1963, Deser, 2014). ROS are clearly controlled by large scale atmospheric circulation (Cohen et al., 2015) emphasizing the need to include internal climate variability uncertainties in the future evolution of ROS studies. The Great Lakes region is one of the area of the world highly impacted by ROS events in winter (Buttle et al., 2016; Cohen et al., 2015). In this region, previous studies found correlations between precipitation and temperature extremes and large-scale circulation indices: The negative phase of the pacific North America oscillation (PNA$^-$) brings more heavy precipitation events in the region south of Great Lakes region (Mallakpour and Villarini, 2016; Thiombiano et al., 2017) and more snowfall in the region North of Great Lakes (Zhao et al., 2013), due to high moisture transport over the region (Mallakpour and Villarini, 2016). Another study showed a negative phase of PNA and positive phase of North Atlantic Oscillation (NAO) associated with warm days (Ning and Bradley, 2015). Temperature and precipitation uncertainties associated to climate internal variability have also been assessed in North America using a global climate model large ensemble (GCM-LE) (Deser et al., 2014). These studies generally separate precipitation and temperature while studying compound events, such as ROS, has been preconized recently to improve our understanding of extreme impacts (Leonard et al., 2014). The





definition of ROS index is also subjected to high uncertainties (Kudo et al., 2017) and this index may not be
relevant in regions affected by decrease of snowpack (Il Jeong and Sushama, 2018). These results emphasize the
need of new compound climate indices to understand the impact of atmospheric circulation on
hydrometeorological extreme events in the Great lake region. In this study, CRCM5-LE, a 50-member regional
model ensemble at a 12km resolution produced over northeastern North America, will be used with the following
objectives:

(1) Define a regional precipitation and temperature compound index that contributes to winter high flows in
Southern Ontario, which is the most populated area in the Great Lakes region.
(2) Assess the relationship between the occurrence of this index and the past large-scale atmospheric circulation.
(3) Investigate the pertinence of the index to explain the future evolution of projected high flows and
(4) Demonstrate how internal variability of climate will modulate the future evolution of atmospheric circulation
and number of hydrometeorological extreme events in the region.
**2 Data and methods**
**2.1 Climate data**
Observations of precipitation, minimum temperature and maximum temperature for the winter months (DJF) in
the 1957-2012 period were taken from the gridded historical weather station data (CanGRD) produced by
McKenney *et al.*, (2011). These data were generated from an interpolation of Natural Resources Canada and
Environment and Climate Change Canada (ECCC) data archives at 10 km spatial resolution. The simulated
evolution of precipitation and temperature are from the Canadian Regional Climate Model Large Ensemble
(CRCM5-LE). CRCM5-LE is a 50-member regional model ensemble at 12km resolution produced over
northeastern North America in the scope of the Québec-Bavaria international collaboration on climate change
(ClimEx project; Leduc et al., 2019). CRCM5-LE is the downscaled version of the global Canadian model large
ensemble (CanESM2-LE) at 310km resolution and offers the possibility to relate each member of CRCM5-LE to
its corresponding member in CanESM2-LE. The future climate data have been bias corrected following the
method of Ines and Hansen (2006) and using the observations and CRCM5-LE historical data in the 1957-2012
period. For each month of the year, the intensity distribution of temperature was corrected using a normal
distribution while the precipitation frequency and intensity distribution were corrected with a gamma distribution.
Each CRCM5-LE grid point has been bias corrected using the closest CanGRD point. Using a unique CanGRD



point for each CRCM5-LE point is permitted in our study because of low elevation gradients between points, the
spatial variability of temperature and precipitation being more dependent on the proximity of the Lakes than the
elevation (Scott and Huff, 1996).

## 2.2 Heavy rain and warm index

Streamflow observations from three watersheds in southern Ontario (Figure 1) were used to define the daily
temperature and precipitation threshold needed to generate high flows in winter. A high flow event was defined
for each watershed as streamflow higher than a threshold equal to the mean streamflow plus three times the
standard deviation. When more than two days in a row were higher than the high flow threshold, the event was
considered as a single event and only the day with the highest high flow was considered. Figure 2 shows for each
high flow event the distribution of daily temperature and precipitation amounts from all grids of the watersheds.
The precipitation and temperature data are from the day situated two days before the high flow event for Big
Creek watershed and three days before the high flow event for Thames and Grand rivers. This lag corresponds to
the delay between a rainfall and/or warm event and the peak flow at the outlet. Figure 2 shows a maximum
temperature higher than 5°C and precipitation higher than 10mm for most grid points during the high flow events.
These temperature and precipitation thresholds are used to create the new index and define days with a significant
rain and warm event that has the potential to generate a high flow event. The 5°C threshold gives a strong
indication that precipitation is in a form of rain, and that the snow in the ground is melting. This index is similar
to the Rain on Snow index (ROS) defined by previous studies. The threshold of 10 mm was previously used to
define ROS events with floods potential (Cohen et al., 2015; Musselman et al., 2018). Our newly created index
can be defined rather as a heavy rain and warm index because snowpack is not integrated in the calculation.

## 2.3 Hydrological modelling

The future evolution of high flows in the three watersheds have been simulated using the Precipitation Runoff
Modeling System (PRMS). PRMS is a semi distributed conceptual hydrological model widely used in snow
dominated regions (Dressler et al., 2006; Liao and Zhuang, 2017; Mastin et al., 2011; Surfleet et al., 2012; Teng
et al., 2017, 2018). PRMS computes the water flowing between hydrological reservoirs (plan canopy interception,
snowpack, soil zone, subsurface) for each hydrological response unit (HRU). For a general description of PRMS
the reader is referred to Markstrom *et al.*, (2015). Champagne *et al.*, (2019) previously applied PRMS to these
three watersheds and extensively described the parametrization process. PRMS has been calibrated in the 1989-
2009 period using Precipitation, minimum temperature and maximun temperature from CanGRD. The three step





trial-and-error calibration approach applied to each watershed showed satisfactory results (Champagne *et al.*,
2019). The streamflow was simulated for each member of the ensemble in the 1957-2055 period using CMIP5-
LE bias corrected data described in the section 2.1.
**2.4 Atmospheric circulation patterns**
The recurrent atmospheric patterns in northeastern North-America were identified by a weather regimes technique
computed by a k-means algorithm (Michelangeli et al., 1995). The algorithm used daily geopotential height
anomalies at 500hPa level (Z500) from the 20[th] century reanalyses (20thCR, Compo *et al.*, 2011) and was applied
in the 1957-2012 period to the northeastern part of North America (30 N-60 N/110 W-50 W). Prior to the k-means
calculations we identified the principal components of the Z500 maps that explain 80% of the spatial variance.
These principal components have been decomposed in weather regimes thanks to the k-means algorithm.  k-means
creates the classes by an iteration method that minimizes intra-regime Euclidean distance and maximize inter–
regime Euclidean distance between the principal components of each day. The algorithm is repeated 100 times
for each number of class between 2 and 10. The choice of the final class number is decided by a red noise test.
This test consists in assessing the significance of the decomposition against weather regimes calculated from 100
randomly generated theoretical datasets that have the same statistical properties than the original dataset. The
weather regimes have been previously calculated for the same domain and the red noise test shows five classes as
the most robust choice (Champagne et al., 2019).

The principal components corresponding to these five classes calculated with 20thCR have been used by the k-
means algorithm to create similarly five weather regimes for each member of CanESM2-LE. The weather regimes
calculated with CanESM2-LE have been calculated in two periods of similar length 1957-2012 and 2013-2068.
Z500 anomalies from CanESM2-LE have been calculated separately for these two periods to avoid a large climate
change signal in the evolution of regime occurrences. The variability of regime occurrences due to internal
variability of climate is therefore fully preserved.
**3 Results**
**3.1 Weather regimes in northeastern North America**
Five weather regimes have been identified in northeastern North America according to the red noise test and show
distinct weather patterns (Figure 3).  The weather regimes computed with 20thCR data show two clear opposite





patterns characterized by positive (HP) and negative (LP) geopotential height anomalies on the Great Lakes. The
regime South was characterized by positive Z500 anomalies in the Atlantic Ocean and negative anomalies in the
north-west part of the domain and was associated with southerly winds. The regime North-West had low
geopotential height on the Gulf of Saint-Lawrence together with winds from the northwest over the Great Lakes
region. Finally, the regime North-East was associated with low geopotential height in the Atlantic Ocean but high
geopotential height close to the Arctic that drove northeastern winds over the Great Lakes. The weather regimes
calculated with CanESM2-LE data in the historical period have very similar patterns (Figure 3). CanESM2 50
members average Z500 anomalies were generally less strong than the 20thCR weather regimes and the anomalies
were slightly shifted to the South. The Z500 anomalies over the Great Lakes were similar for most of the regime
except for regime South showing higher Z500 anomalies.

## 3.2 Validation of heavy rain and warm index and high flows simulated by CRCM5-LE

The ability of CRCM5-LE to simulate the occurrence of heavy rain and warm events is assessed in this section.
The number of heavy precipitation events per winter was generally well recreated by the regional ensemble in the
historical period (Figure 4). These events were generally overestimated in the north and western parts of the region
especially in areas close to Lake Huron. In this region, few grid points show all 50 members of the ensemble
overestimating the number of events compared to the observations. The number of warm events followed a similar
spatial variability with more frequent events in the southern parts, particularly in the Niagara peninsula between
Lake Erie and Lake Ontario. The number of warm events was overestimated by all members in the entire area
except for the Lake Simcoe Area (Figure 4 Centre-right plot in blue). The number of compound events, heavy
rain and warm temperature was more frequent in the area close to Lake Erie in both observations and simulations.
The number of events was overestimated by the ensemble mean in the northern parts of the region. In this region,
many grid points show all members of the ensemble overestimating the number of events. Close to Lake Erie the
overestimation was lower and even non-existent in the Niagara peninsula.

The ability of the ensemble to recreate the number of heavy rain and warm events relative to the number of
occurrences of each weather regime has been assessed for the heavy precipitation events, the warm events and the
compound events. For the heavy precipitation events the observations show high number of events during the
occurrence of regimes HP in the southern parts of the region and especially in inland areas (Figure 5). The regime
South show the second largest number of heavy precipitation events while the regime North-West was associated
with the least number of heavy precipitation events. The number of precipitation events associated to a regime LP




is spatially variable with a large number of events limited to the Lake Huron shoreline. The ensemble appeared to
recreate with accuracy these number of events per weather regimes. The regime South is the exception with almost
twice more events per occurrence of regime with the 50 members average compared to OBS. In southern areas
the simulations were also slightly overestimating the number of heavy precipitation events during regime North-
West while underestimating during regime HP (Figure 5).

Concerning the observed warm events, they also occurred mostly during regime HP while they were non-existent
during regime LP (Figure 6). The number of warm events was similar between regimes North-West, North-East
and South in a large part of the area. In the Niagara peninsula more events were occurring during a regime South.
The simulations recreated well the number of warm events for the regimes HP, LP and North-East while it
overestimated the number of events for the two other regimes and especially the regime South (Figure 6). The
number of events per occurrence of regime South for the 50 members average was twice the number of events
calculated with the observations.

The compound index heavy rain and warm events was also more frequent during a regime HP in both observations
and simulations while the occurrences of events were very low for LP and North-West (Figure 7). The simulations
overestimated these events by 3 to 4 times for the regime South while it was well recreated for the other regimes
(Figure 7).

The historical distribution of streamflow associated to heavy rain and warm events for the observed streamflow
(OBS), streamflow simulated with CanGRD (CTL) and streamflow simulated with each CRCM5-LE member
(ENS) is shown is Figure 8. The results show an observed streamflow frequently higher than the high flows
threshold when the heavy rain and warm events occurred during a regime HP. Few days also show high flows
during a regime South especially in Thames River and Big Creek watersheds. The streamflow simulated with
CanGRD weather data (CTL) is underestimated but show a similar inter-regime variability with higher streamflow
during HP heavy rain and warm events compared to events associated with other weather regimes. The 50
simulations from CRCM5-LE show also higher streamflow when heavy rain and warm events correspond to
regimes HP or South (Figure 8).





### 3.3 Future evolution in the number of hydrometeorological extreme events


The total number of heavy precipitation events simulated by CRCM5-LE is expected to increase between 1961-
1990 and 2026-2055, with a maximum increase between 1 and 2 events per winter expected close to the Georgian
Bay (Figure 9). The increase in the number of events is mainly expected during the regime South but also for the
regime LP near Lake Huron and HP between the Georgian Bay and Lake Ontario. The increased frequency of
warm events is expected to be even higher reaching a total increase of about 10 events close to Lake Erie. The
highest increase is expected for HP regime and at a lower rate for regimes South and North-West. The number of
compound events is expected to increase by 1 or 2 events per winter with a maximal increase between Lake Erie
and Huron. The increase in the number of heavy rain and warm events is expected to concern mainly the regime
South and HP (Figure 9).

The contribution of the trend in heavy rain and warm events to the trend in number of high flows has been
investigated (Figure 10). For each member of the ensemble, the historical number of high flows events associated
to each weather regime has been multiplied by the change factor between number of Rain and warm events in the
historical period and in the future period. The difference between this calculated number of high flows and the
historical number of high flows corresponds to the theoretical high flows frequency change due to the occurrence
change in number of heavy rain and warm events (OCC). The total change in number of high flows (TOT)
corresponding to each weather regimes is subtracted by OCC for each ensemble member to account for a change
in number of high flows not due to a change in number of heavy rain and warm events (DIF). Taking all weather
regimes events together, TOT is expected to increase in the future. The increase in OCC is similar to the increase
in TOT even though OCC is slightly higher than TOT in the Big Creek watershed.  Considering HP's events only,
the increase of OCC is higher than TOT while for events associated with regime South TOT is higher than OCC.

### 3.4 Relationship between change in occurrence of weather regimes and extreme events


Correlations between change of occurrence of weather regimes and change in number of Rain and Warm events
between 1961-1990 and 2026-2055 for the 50 members have been calculated for each grid point (Figure 11). The
correlations between occurrence of weather regimes and warm events is higher compared to correlations with
heavy precipitation events. The results show positive correlations between warm events and regime HP and
negative correlations between warm events and regime LP/North-east in the entire area. The change in number of
warm events is also positively correlated to the change in occurrence of regime South but the results are not
significance (95% confidence). The correlations with the compound index are less spatially spread with positive





correlations between the index and the regime HP close to Lake Erie and negative correlations with regime LP
near Lake Huron.

Correlations between combination of weather regimes change and Rain and warm index change averaged over
the entire region have also been investigated (Table 1). The combinations of weather regimes have been done by
summing the change of occurrence from the two regimes of each combination. The correlations between change
of any weather regimes combinations and change in number of heavy precipitation events are not significant. The
correlations between change of number of warm events and change in occurrence of weather regimes is improved
when regime South is associated to regime HP and when regime LP is associated to regime NE compared to
correlations with regimes HP or LP only (Table 1). Concerning the heavy rain and warm index the correlations
are not significant if the regimes HP and South are correlated separately to the number of events but are positive
and significant (95% confidence interval) if the correlation is applied to a combination of regime South-HP and
negative and significant (90% confidence interval) with a combination of North-west-LP. The correlation with
the high flows in each of the three watersheds have also been investigated (Table 2) and shows significance only
in the Big Creek watershed. A combination of HP-LP is negatively correlated to high flows while North-west and
a combination North-west-South are positively correlated to high flows.

The change of heavy precipitation, warm and compound events frequency in respect to change of occurrence of
regimes South, HP, LP and North-west for each member of the ensemble is shown in Figure 12. The
correspondence between change in number of heavy precipitations events and change in number of occurrences
of weather regimes is not clear, confirming the low correlations in Figure 11 and Table 1. Regarding the warm
events, the large increase in occurrence of regime HP-South or large decrease in regimes LP-North-West are
generally associated to a large increase in number of warm events confirming the results from Figure 11 and Table
1. Concerning the compound index, despite the correlations shown in Figure 11 and Table 1, a high increase of
HP and South occurrences does not systematically lead to a large increase in number of events (Figure 12).
**4 Discussion**
**4.1 Atmospheric circulation and extreme weather events**
The results show that the occurrence of heavy rain and warm events are modulated by specific atmospheric
patterns in winter which corroborates previous studies in the Great Lakes region. These studies found that heavy
precipitation and flooding events are associated to high geopotential height anomalies in the east coast of North
America similarly to regimes HP or South (Mallakpour and Villarini, 2016; Zhang and Villarini, 2019; Farnham
*et al.*, 2018)). Our results found differences between observations and simulations with more heavy precipitation
events during regime HP in the observations while the simulations with CRCM5-LE show more precipitations
events during regime South (Figure 5). The overestimation of the number of precipitation events for regime South
can be associated to the difference in pattern between regimes calculated with 20thCR and CanESM2-LE (Figure
3). Regime South calculated with CanESM2-LE shows Z500 anomalies shifted to the west and likely a more
meridional flux compared to the regime South from 20thCR. The weather regimes associated to heavy
precipitations in the Mid-west defined by Zhang and Villarini, (2019) show high pressure anomalies on the east
and low pressure on the west sides of the Great lakes similarly to regime South calculated with CanESM2. The
regime South calculated with 20thCR show negative Z500 anomalies with a northern position compared to
CanESM2-LE and therefore a stronger zonal flux while the regime South calculated with CanESM2-LE has likely
a more meridional flux driving humidity from the Gulf of Mexico (Figure 3). This pattern also brings warm
temperature events even though the regime HP brings even more warm events in both the observations and the
ensemble average (Figure 6). Regime HP has similarities with the positive phase of the NAO clearly associated
with warm winter temperature in the Great Lakes region (Ning and Bradley, 2015). The other weather regimes
bring generally fewer heavy precipitation or warm events apart from regime LP bringing heavy precipitation close
to Lake Huron (Figure 5). LP is not associated with warm events (Figure 6) suggesting that these extreme
precipitations are in form of snow and likely from lake effect snow. Suriano and Leathers, (2017) show that low
pressure anomalies north-east from Great lakes brings major lake effects snow in the eastern shores of Lake Huron
due to less zonal wind and cold outbreaks from the Arctic. The regime LP shows low geopotential height right on
the Great lakes and the associated north-west winds on the Lake Huron are likely to bring lake effect snowfall in
this area.
**4.2 Future evolution of rain and warm events**
The future increase of heavy precipitations events in winter in Southern Ontario was already described in Deng *et*
*al.*, (2016). Compound events such as Rain on Snow (ROS) events have also been investigated by Il Jeong and
Sushama (2018). These authors defined ROS events as liquid precipitation and snow cover higher than 1mm and
found no significant trend of ROS events in the Great lake region, in continuity to what was observed in the past
(Wachowicz et al., 2019). These studies show that the Great Lakes region is located between a region of increase
ROS events due to increase of rainfall in the north and a decrease in ROS events due to decrease of snowpack in



southern regions. Increase of rainfall and decrease of snowpack are both likely to occur in Southern Ontario and
are cancelling each other in term of ROS events. Our study does not consider snowpack and show an increase in
heavy rain and warm compound events (Figure 9). The increase of heavy rain and warm events is likely driven
by warmer temperature shown by the increase of the compound events and warm events both occurring at a higher
extent close to Lake Erie (Figure 9). The increase in extreme precipitation events is less significant than the
increase of warm events and is occurring mostly in the Northern parts of the area (Figure 9).

The future evolution of ROS or heavy rain and warm events corresponding to different weather patterns have not
been yet investigated in previous literature. It is interesting to note that the future increase of the rain and warm
events are expected to occur only for the regimes HP and South, the number of events remaining very low for the
other regimes (Figure 9). This result suggests that the global increase of mean temperature and precipitation is not
sufficient to reach the 10 mm and 5°C threshold for LP, North-West and North-East regimes. More precipitation
events are expected during regimes LP but the temperature stays too low to increase the numbers of heavy rain
and warm events (Figure 9). Regime North-West shows an increase of warm events but not an increase in
precipitation events and therefore the number of rain and warm events is not expected to increase.
**4.3 Change in frequency of heavy rain and warm events partially modulated by the occurrence of weather**
**regimes**
Despite clear association between regimes HP/South and occurrences of rain and warm events, the uncertainties
linked to internal variability of climate are not fully apprehended by the frequency of weather regimes. Members
of the ensemble associated with a simultaneous high increase of regime HP and South frequencies are generally
associated with higher increase in rainfall and warm events (Table 1) but the association is less straightforward
than suggested by the correlation values (Figure 11) probably due to poor association between precipitation
extremes and occurrence of weather regimes (Table 1 and Figure 10). Similar change in occurrences of South-HP
weather regimes can lead to variable change in number of heavy rain and warm events (Figure 11). This suggests
that other scales than the weather regimes calculated in the northeastern North American domain are likely to play
a role in weather extreme events and especially the change of heavy rain and warm events and precipitation events.
The presence of the Great lakes has a large role in the variability of precipitation at a local scale (Martynov et al.,
2012) suggesting that variability of precipitation events depend not so much on the atmospheric circulation over
the Great Lakes at the day of the events. The temperature of the lakes and the amount of ice covering the lakes
plays a great role in the variability of precipitation (Martynov et al., 2012).



### 4.4 Non stationarity in the relationship between weather extreme events and high flows

The projections show that the increase in number of high flows associated to a regime HP is expected to be lower than the increase in number of heavy rain and warm events (negative DIF in Figure 10). This result suggests that the conditions to produce high flows may change in the future. As the temperature increase, snowmelt is expected to be a less important component in the generation of high flows in the region. In the historical period regimes HP and South produce approximately the same number of high flows in the simulations (Figure 7) but are driving mostly by heavy precipitation for the regime South and warm events for the regime HP (Figure 5 and 6). More importantly, HP shows a further increase of warm events in the future while South show rather an increase of precipitation (Figure 9). In the context of less snow, the importance of precipitation to drive high flows will be higher in the future because warmer conditions do not increase snowmelt in case of a snowpack reduction. Therefore, the increase of weather extreme events associated to the regime South will be associated to an increase of high flows more strenuously than the increase of events associated to HP.

The future change in number of high flows is associate to a large inter-member uncertainty (Figure 10). The weather extreme events inter-member uncertainty was partly associated to the change in occurrence of weather regimes especially for the warm component (Figure 11 and 12 and Table 1). The association between occurrence of weather regimes and high flows is less clear and shows opposite results (Table 1 and 2). Especially, change of occurrence of regime North-west is positively correlated to the change in number of high flows in Big Creek watershed (Table 2) while it is negatively correlated to the change in number of weather extreme events in this area (Figure 11). The correlation is also significant when regimes North-west and South are associated (Table 2). This result can be due to the continuous nature of streamflow and the preferential sequence of weather regimes. Regime North-west shows an increase in number of warm events especially close to Lake Erie (Figure 9) with the potentiality to melt more snow in the future. The amount of precipitation generated by a regime North-west is probably not sufficient to generate high flows (Figure 9), but the increase of snowmelt during the regime North-West likely enhances streamflow that make the high flows threshold easier to reach in a following precipitation event. The pattern associated with regime North-west shows anticyclonic systems in the west part of the domain (Figure 3). The meteorological systems have a tendency to move eastward and this anticyclonic system is likely to become a regime South or HP (Champagne *et al.*, 2019, Supplementary material, Table S2). As already stated in the previous paragraph, regime HP will be less likely to produce a heavy rain event than a regime South in the future. The combination of the warmer regime North-west following by a wetter and warmer regime South are therefore more likely to produce high flows in the future. These results emphasize the need to study not only each



hydrometeorological extreme events and relationship with atmospheric circulation independently, but also
focusing on the sequence of weather patterns preceding the high flows events.

### 4.5 Relevance of rain and warm events to explain future evolution of high flows

One of the objectives of this study was mainly to create a new index that explains high flows in Southern Ontario
and investigate how this index will change in the near future. However, as stated in the previous section, the
relationship between the extreme weather events index and high flows is affected by non-stationarity. Applied in
the past, the Rain and warm index works well to define the high flows risk in Southern Ontario (Figure 2), the
warm component of this index being a condition to trigger snowmelt. In a warming climate, snowpack is reduced,
and the rain to snow ratio is increasing (Il Jeong and Sushama, 2018), changing the relationship between extreme
weather events and high flows. Rain on snow index could be used in lieu of our heavy rain and warm index but
this index is not projected to be more frequent in the future in the Great Lakes region, precisely because of less
snow in the ground (Il Jeong and Sushama, 2018). Moreover, ROS index integrate events with a very small
contribution of snowmelt to the high flows while neglecting rainfall only events (Cohen et al., 2015; Il Jeong and
Sushama, 2018; Pradhanang et al., 2013). The definition of ROS also introduces more uncertainties as it depends
on the combination of simulated precipitation and temperature for several days (Kudo et al., 2017). Our heavy
rain and warm index minimizes this uncertainty and take into consideration heavy rainfall whatever the amount
of snow covering the ground. It is therefore a good tool to assess the potential risk of high flows in Southern
Ontario from all ranges of rain events, even though it is important to keep in mind that the flood risk diminished
as snowpack decreases. A rain only index could also be used but the impact of snowpack on streamflow would be
completely eradicated while snow will still play a role in the future hydrology. ROS events, liquid precipitation
events and our heavy rain and warm events should be investigated together to fully understand the future evolution
of the flood risk due to a shift in weather extreme events.

### 5 Conclusion

The aim of this study was to assess the ability of the Canadian Regional Climate Model Large Ensemble (CRCM5-
LE), a downscaled version of the 50-members global Canadian model Large Ensemble (CanESM2-LE), to
simulate winter hydrometeorological extreme events in Southern Ontario and to investigate how the internal
variability of climate will modulate the future evolution of these extremes. The winter composite index heavy rain
and warm temperature was identified in the past with gridded observation data (CanGRD) by investigating what





conditions of temperature and precipitation are necessary to produce a high flow in three watersheds in Southern
Ontario. PRMS model was used to simulate the future evolution of high flows for each member of CRCM-LE in
these three watersheds. The large-scale circulation patterns corresponding to these events were assessed by
identifying past recurrent weather regimes based on daily Z500 from the 20th century reanalyses and estimating
the evolution of the same weather regimes in the future for each member of CanESM2-LE. The results of this
study show that CRCM5-LE was able to:
(1)  Recreate the historical larger number of events close to Lake Erie despite an overestimation of warm
388         events.

(2)  Simulate more heavy rain and warm events as well as high flows during the regimes associated with high
390         pressure anomalies on the Great Lakes (HP) and the Atlantic-Ocean (South).

(3)  Project an increase in the future number of heavy rain and warm events and associated high flows
392         especially during the regimes HP and South and in the vicinity of Lake Erie.

These results suggest that depending on the future evolution of natural variability of climate, the increase in the
number of events will be amplified or attenuated by the favoured positions of the pressure systems. The natural
variability of climate is not expected to greatly modulate the number of high flows due to an increase of the
importance of precipitation in generating high flows. The role of more localized processes such as impact of the
lakes on precipitation events needs to be further evaluated to improve the ability of the next versions of regional
climate models to recreate the precipitation events. The newly created weather index did not integrate snowpack
because the uncertainties in the ability of CRCM5-LE to recreate precipitation and temperature extremes at a daily
basis would be further increase in snowmelt estimates. However, snowpack variability will have a large impact in
the modulation of high flows in the region and future studies should investigate snow processes by taking
advantage of rapid improvements in climate regional modelling. Other regional climate models and different
scenarios should also be used to improve our understanding in the future evolution of hydrometeorological
extreme events in Southern Ontario. Despite these future possible improvements, our study gives a good
estimation of what to expect in term of change in number of hydrometeorological events in Southern Ontario and
will serve to better estimate the future flood risk in this populated region.

**Authors contribution**

ML furnished CRCM5-LE data. OC performed the analyses and made the figures. OC prepared the manuscript
with contributions from all co-authors.



**Competing interest**
The authors declare that they have no conflict of interest.
**Acknowledgement**
Financial support for this study was provided by the Natural Sciences and Engineering Research Council
(NSERC) of Canada through the FloodNet Project. We also acknowledge support and contributions from Global
Water Future Program, Environment and Climate Change Canada, Natural Resources Canada and Water Survey
of Canada. The production of ClimEx was funded within the ClimEx project by the Bavarian State Ministry for
the Environment and Consumer Protection.The CRCM5 was developed by the ESCER centre of Université du
Québec à Montréal (UQAM; www.escer.uqam.ca) in collaboration with Environment and Climate Change
Canada.We acknowledge Environment and Climate Change Canada's Canadian Centre for Climate Modelling
and Analysis for executing and making available the CanESM2 Large Ensemble simulations used in this study,
and the Canadian Sea Ice and Snow Evolution Network for proposing the simulations. Computations with the
CRCM5 for the ClimEx project were made on the SuperMUC supercomputer at Leibniz Supercomputing Centre
(LRZ) of the Bavarian Academy of Sciences and Humanities. The operation of this supercomputer is funded via
the Gauss Centre for Supercomputing (GCS) by the German Federal Ministry of Education and Research and the
Bavarian State Ministry of Education, Science and the Arts.

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









**Figure 1: Location of the three watersheds and the ClimEx grid points used in this study and situation in the northeastern North America domain (Inset)**




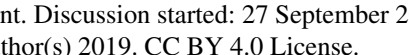

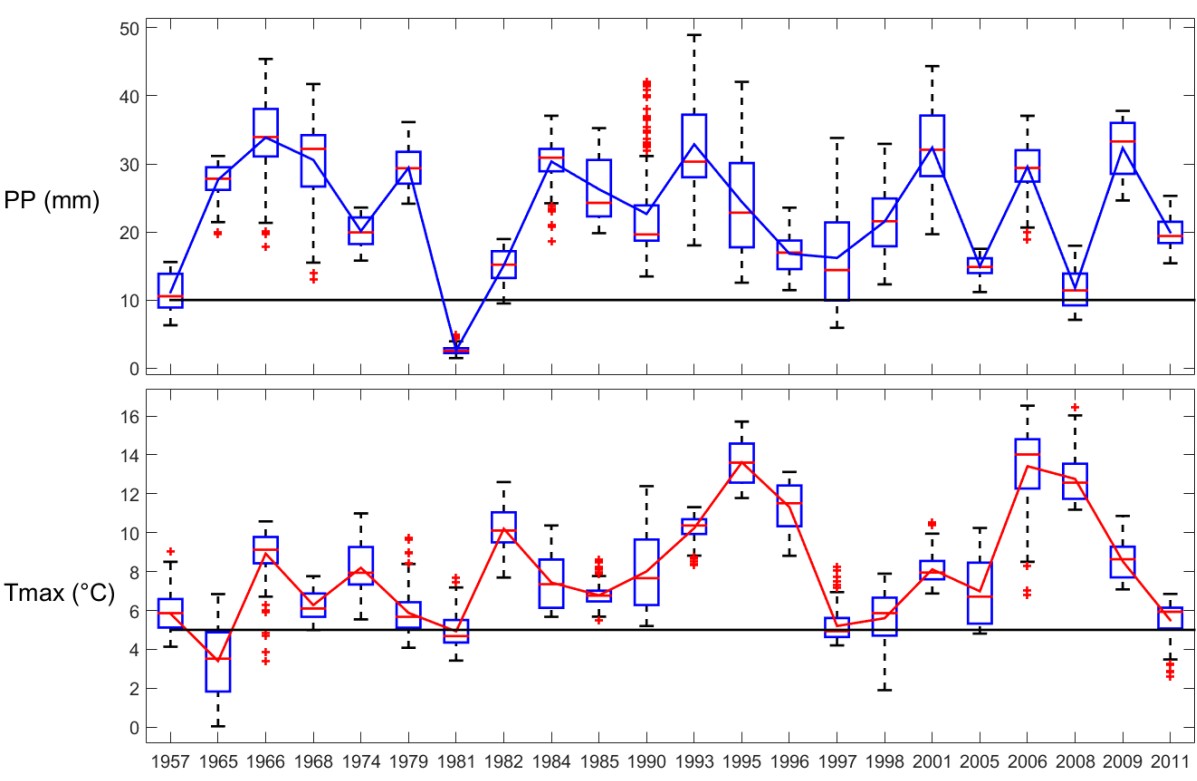


**Figure 2: Distribution of CanGRD temperature and precipitation from all 3 watersheds grid-points corresponding to each DJF high-flow event. Boxes extend from the 25th to the 75th percentile, with a horizontal red bar showing the median value. The whiskers are lines extending from each end of the box to the 1.5 interquartile range. Plus signs correspond to outliers. The blue lines correspond to high flows (Average streamflow plus 3 times the standard deviation). The horizontal black lines correspond to the thresholds used to define DJF extreme events.**

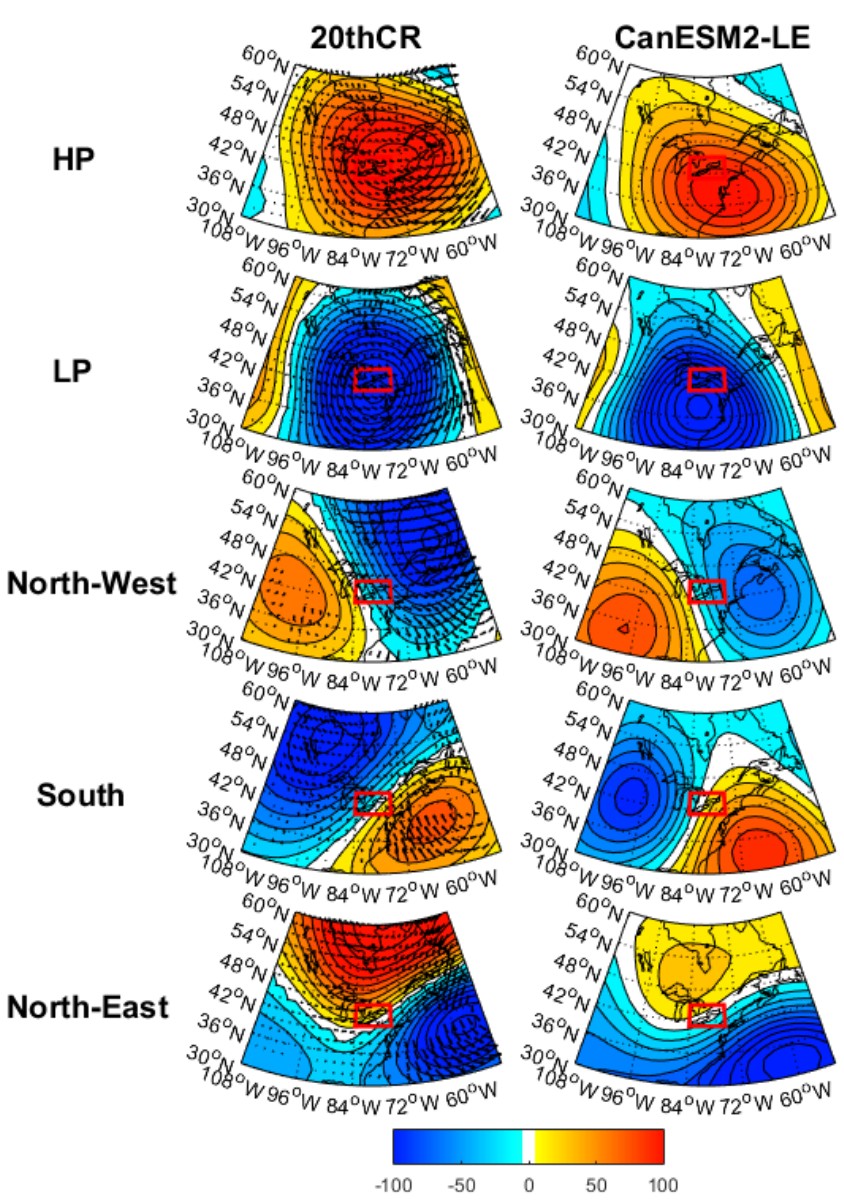

**Figure 3: Left: DJF Z500 anomalies (colours) and winds (vectors) corresponding to Weather regimes calculated with 20thCR. Right: DJF 50 members average Z500 anomalies calculated with CanESM2-LE.**



**Figure 4: DJF number of precipitation and warm extreme events in the historical period (1961-1990) for CanGRD (left**
**panels), 50 members CRCM5-LE average (mid panels) and CanGRD minus CRCM5-LE (right panels). The dotted**
**lines in the mid panels represent the standard deviation of the 50-members CRCM5-LE simulated number of events.**
**Stippled regions in the right panels indicate where the observations lie within the CRCM5-LE ensemble spread.**

**Figure 5: Percentage of DJF number of precipitation events relative to DJF occurrence of weather regimes in the historical period (1961-1990) for CanGRD (upper panels), 50 members CRCM5-LE average (mid panels) and CanGRD minus CRCM5-LE (right panels). The dotted lines in the mid panels represent the standard deviation of the 50-members CRCM5-LE simulated percentage. Stippled regions in the lower panels indicate where the observations lie within the CRCM5-LE ensemble spread.**



574

**Figure 6: Percentage of DJF number of warm events relative to DJF occurrence of weather regime in the historical period (1961-1990) for CanGRD (upper panels), 50 members CRCM5-LE average (mid panels) and CanGRD minus CRCM5-LE (lower panels). The dotted lines in the mid panels represent the standard deviation of the 50-members CRCM5-LE simulated percentage. Stippled regions in the lower panels indicate where the observations lie within the CRCM5-LE ensemble spread.**



**Figure 7: Percentage of DJF number of heavy rain and warm events relative to DJF occurrence of weather regimes in the historical period (1961-1990) for CanGRD (upper panels), 50 members CRCM5-LE average (mid panels) and CanGRD minus CRCM5-LE (lower panels). The dotted lines in the mid panels represent the standard deviation of the 50-members CRCM5-LE simulated percentage. Stippled regions in the lower panels indicate where the observations lie within the CRCM5-LE ensemble spread.**



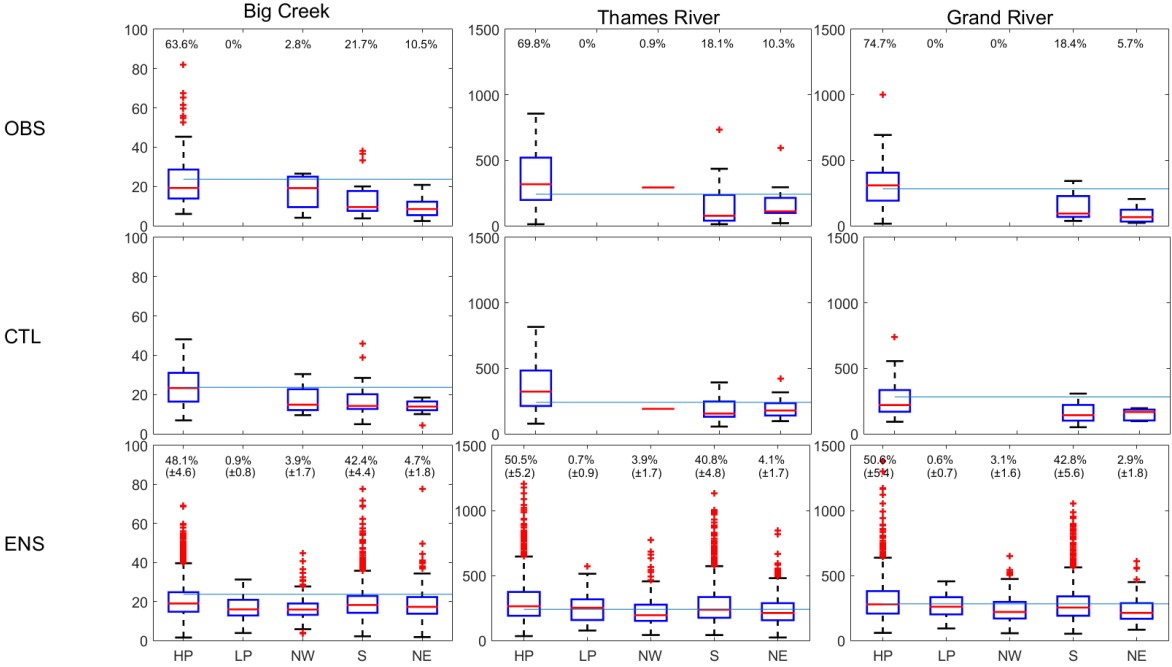

586

**Figure 8: Upper and mid panels: Distribution of observed (OBS) and simulated (CTL) streamflow corresponding to all observed heavy rain and warm events. Lower panels: Distribution of simulated streamflow corresponding to all simulated heavy rain and warm events (ENS). Boxes extend from the 25th to the 75th percentile, with a horizontal red bar showing the median value. The whiskers are lines extending from each end of the box to the 1.5 interquartile range. Plus signs correspond to outliers. The horizontal blue lines correspond to high flows (Average streamflow plus 3 times the standard deviation).**

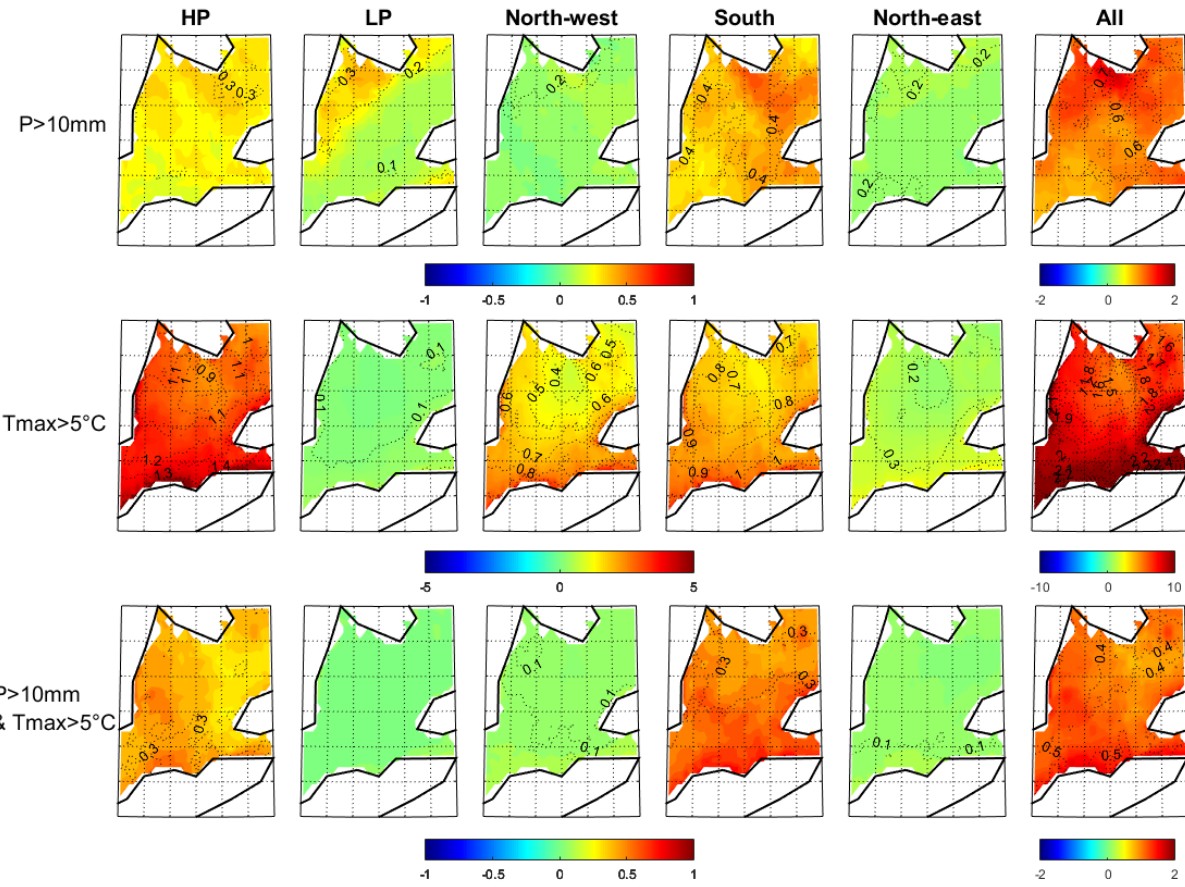

593

**Figure 9: DJF change in number of precipitation and warm events between the historical (1961-1990) and the future period (2026-2055) for the 50 members CRCM5-LE average. The dotted lines represent the standard deviation of the 50-members CRCM5-LE simulated change in number of events.**





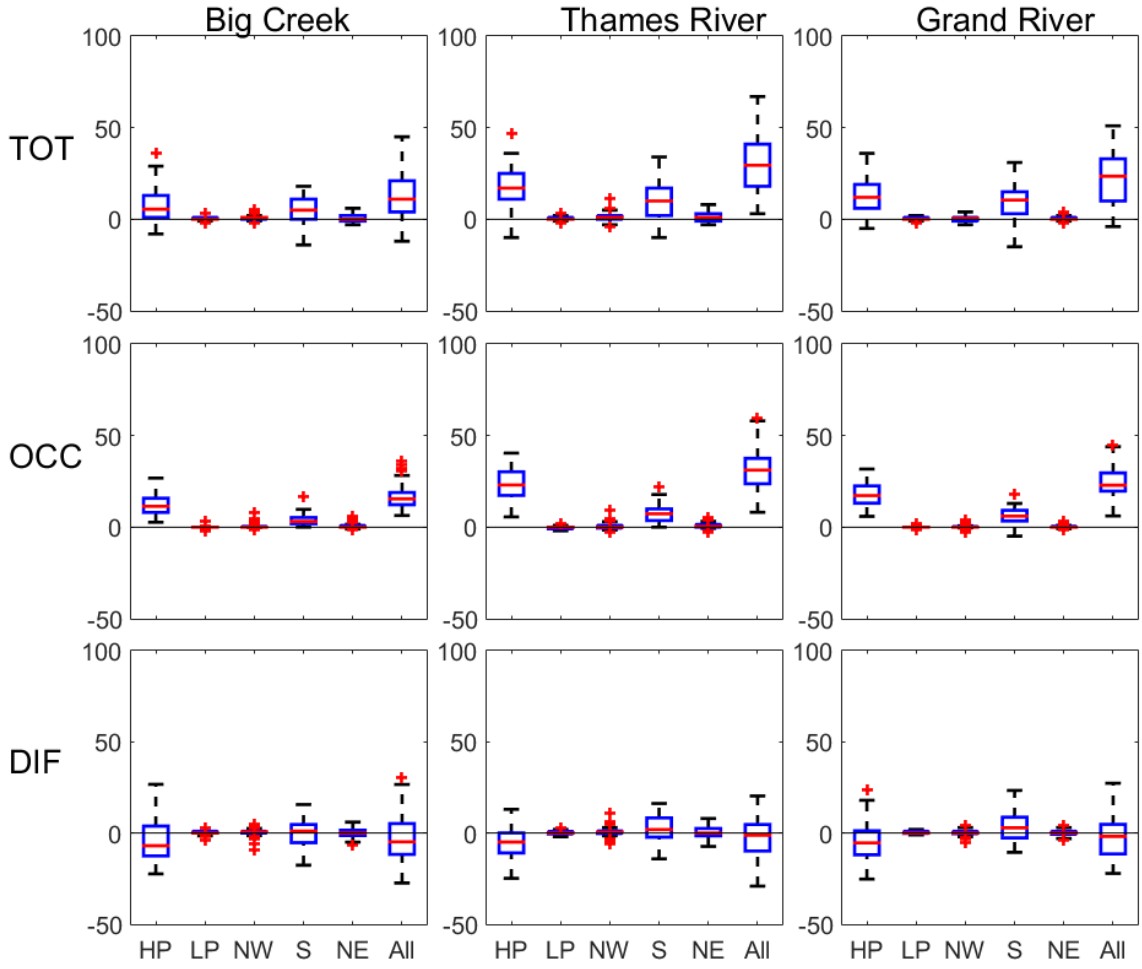

597

**Figure 10: upper panels: Distribution of change in number of high flows between 1961-1990 and 2026-2055 simulated**
**from the 50 members of the ensemble (TOT). Mid panels: Distribution of theoretical change in number of high flows**
**using the factor of change in number of heavy rain and warm events between 1961-1990 and 2026-2055 (OCC). Lower**
**panels: TOT minus OCC (DIF). Boxes extend from the 25th to the 75th percentile, with a horizontal red bar showing**
**the median value. The whiskers are lines extending from each end of the box to the 1.5 interquartile range. Plus signs**
**correspond to outliers.**

**Figure 11: DJF inter-members correlations between change in occurrence of weather regimes and change in number of events between 1961-1990 and 2026-2055. Black points indicate a correlation significant at 95% according the Pearson's correlation table.**












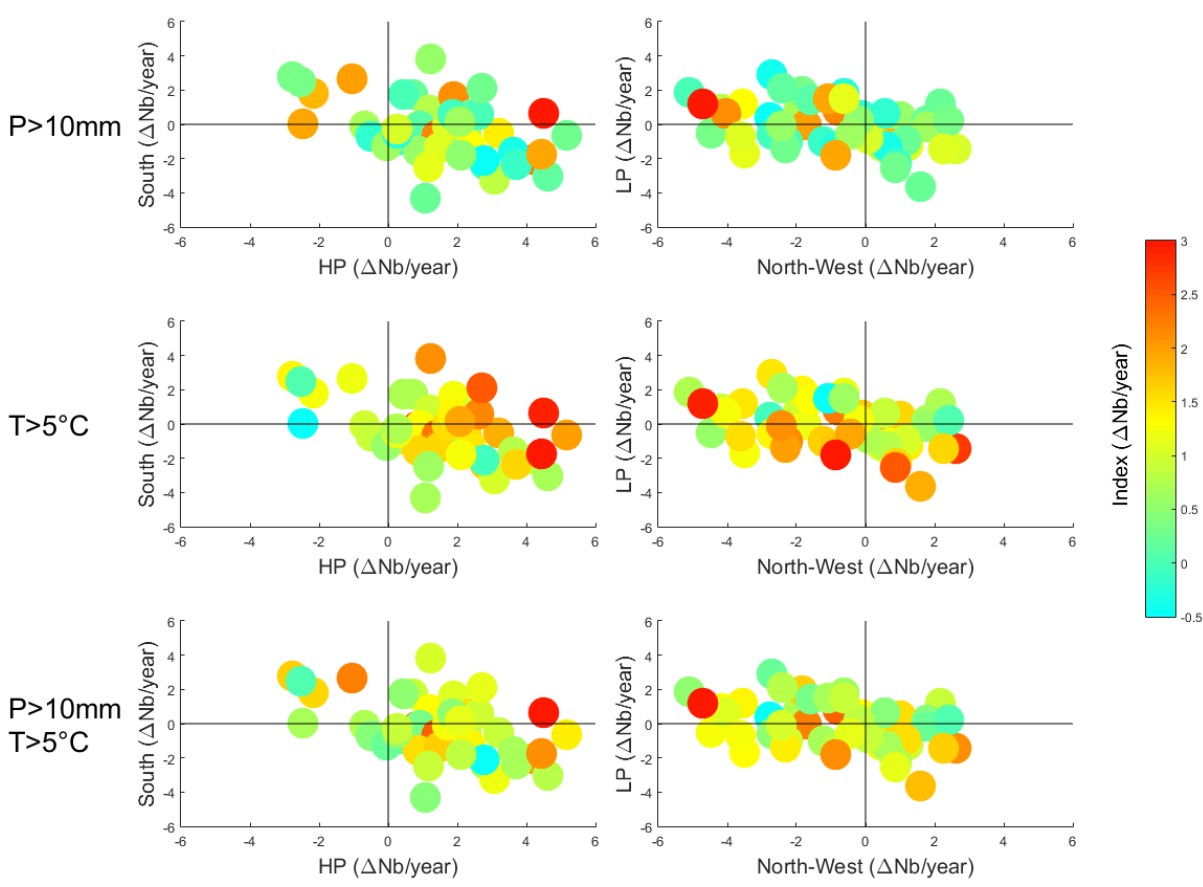

**Figure 12: DJF change in occurences of regimes HP-South (left) and LP-north-West (right) in respect to change in number of precipitation and warm events (Colours) for each member of CRCM5-LE between 1961-1990 and 2026-2055.**





**Table 1: inter-members correlations between DJF change in occurrence of weather regimes and DJF change in number**
**of events between 1961-1990 and 2026-2055. Bold show correlations significant at 90% confidence level, a single**
**underline significant at 95% and double underline significant at 99% according to the Pearson's correlation table.**

|     | P>10mm | | | | | Tmax>5°C | | | | | P>10mm & Tmax>5°C | | | | |
|-----|------|------|------|------|------|------|------|------|------|------|------|------|------|------|------|
|     | HP | LP | NW | S | NE | HP | LP | NW | S | NE | HP | LP | NW | S | NE |
| HP | - 0.01 | -0.01 | -0.15 | 0.09 | 0.06 | **0.43** | 0.15 | **0.32** | **0.55** | 0.12 | 0.21 | 0.05 | 0.08 | **0.35** | 0.08 |
| LP |  | 0 | -0.17 | 0.09 | 0.18 |  | **-0.38** | **-0.26** | -0.13 | **-0.46** |  | -0.21 | **-0.23** | -0.01 | -0.21 |
| NW |  |  | -0.17 | -0.06 | -0.09 |  |  | 0 | 0.09 | **-0.26** |  |  | -0.08 | 0.04 | -0.08 |
| S |  |  |  | 0.12 | 0.18 |  |  |  | 0.13 | -0.16 |  |  |  | 0.16 | 0.04 |
| NE |  |  |  |  | 0.10 |  |  |  |  | **-0.28** |  |  |  |  | -0.10 |


**Table 2: inter-members correlations between DJF change in occurrence of weather regimes and DJF change in number**
**of high flows events between 1961-1990 and 2026-2055. Bold show correlations significant at 90% according to the**
**Pearson's correlation table.**

|     | Big Creek | | | | | Thames River | | | | | Grand River | | | | |
|-----|------|------|------|------|------|------|------|------|------|------|------|------|------|------|------|
|     | HP | LP | NW | S | NE | HP | LP | NW | S | NE | HP | LP | NW | S | NE |
| HP | - 0.13 | **-0.25** | 0.12 | -0.09 | -0.16 | -0.08 | -0.12 | 0.02 | -0.02 | -0.09 | -0.10 | -0.19 | 0 | 0.03 | -0.10 |
| LP |  | -0.18 | 0.15 | -0.08 | -0.16 |  | -0.06 | 0.05 | 0.01 | -0.07 |  | -0.13 | 0 | 0.02 | -0.10 |
| NW |  |  | **0.26** | **0.25** | 0.21 |  |  | 0.09 | 0.12 | 0.06 |  |  | 0.08 | 0.15 | 0.07 |
| S |  |  |  | 0.04 | -0.03 |  |  |  | 0.06 | -0.02 |  |  |  | 0.14 | 0.10 |
| NE |  |  |  |  | -0.08 |  |  |  |  | -0.04 |  |  |  |  | -0.02 |
