# Peer review of "Winter hydrometeorological extreme events modulated by large scale atmospheric circulation in southern Ontario"

_Earth System Dynamics, 2019_

## Referee Comment (RC1) · Anonymous Referee #1 · 7 Nov 2019

This manuscript presents an index to describe warm, heavy rain events during winter in southern Ontario. This index is then used to evaluate the occurrence of such events with different large scale circulation patterns and projections for the future. The paper presents a thorough analysis and I appreciate the detailed discussion included in Section 4. Some concerns are listed below.

My main concern is that the authors have not provided sufficient justification for the need to create a new index. The introduction focuses on rain on snow events, but this new index does not consider snow cover. This disconnect needs to be more fully explained. Why is the proposed index better than those used in previous literature?

[Figure]

Considering this is a special issue on large ensembles, I think there should be more discussion about how this work is taking advantage of the large ensemble used here.

Specific comments:

Line 52: Is the "precipitation" here referring to rain or snow or both?

Line 84: Was only the future period bias corrected?

Line 87: Is precipitation frequency also modeled with a gamma distribution? This makes more sense for intensity.

Line 138-141: What effect would the trends within these two periods have?

Section2.4/Section 3.1: Were the top regimes determined separately for the model and reanalysis? If so, then I think the good agreement warrants more commentary. If not, can you describe more clearly how they are related?

Section 3.2: Is the modelled and bias-corrected data being compared against the dataset used for the bias correction target? If so, can you add a brief discussion on the implications?

Section 3.3: What does the model bias in the South regime mean for these projections? How robust are the streamflow projections when they do not account for changes in snowfall/snowpack or associated feedbacks?

Line 229: Do you mean the magnitude of the correlations here?

Line 237: Can you clarify what is meant by the combination of weather regimes? The value of using these combinations instead of just changes in individual regimes is not clear.

Figure 1: I found it very difficult to orient myself in this figure. It is hard to discern the coastlines of the Great Lakes from the contours in the large scale figure and the legend covers the land separating two lakes. It would be very helpful to label the lakes and

[Figure]

key places here, especially since so much of the later discussion is very specific about local geography.

Figure 2: Why is the scale of the horizontal axis nonlinear? With the current spacing and connecting line, this time series is misleading.

Figures 4-7: The figure label ("simulations minus observations") does not agree with the caption ("CanGRD minus CRCM5-LE")

Figure 8: Are the ensemble values pooled or averaged?

Line 462: This paper should be cited as Jeong and Sushama

———————————————————

---

## Referee Comment (RC2) · Anonymous Referee #2 · 8 Nov 2019

The authors investigate the occurrence of rain on snow events with a single model ensemble. Overall this is a very interesting paper with a good application of the bottom-up approach that has recently been endorsed for studying compound events.

Using the compound exceedance of a precipitation and temperature threshold seems relatively naive, given that large increases in temperature in the future will lead to very different snow cover patterns, a key determinant of ROS events. A more appropriate variable than temperature seems to me the difference in surface snow amount between consecutive days, which could be used as a proxy for snowmelt. This is available from the model output. With this it should be possible to build a better compound index that

should also be more reliable in future projections.

I suspect streamflow is very non-gaussian distributed. In particular, it's asymmetric and bounded from below. Taking the mean +3 standard deviations as an indicator for extremes is thus very unintuitive and not really appropriate for such a distribution. I would suggest to use a high percentile (e.g. above the 99th percentile, or something similar, could also be more extreme). This can then also be translated easily into a return period.

Would it be an option to us only the weather patterns based on the observations and classify the models according to those? This might reduce differences between models and observations with respect to the occurrence rate of heavy precip and warm events (the authors discuss this point in sec 4.1).

Please mention somewhere explicitly how the compound index is defined. Is it just the occurrence of events where both temperature and precipitation exceed a certain threshold? Or the number of such occurrences?

Minor comments: I would recommend the authors to do a thorough spell check and grammar check. There are a number of minor grammatical errors and typos in the text.

L 49: start new paragraph

L59: "preconized" ?

L67: "contributes to": maybe better: "explains the variability of"

L69: "occurrence of the index": an index does not occur, it has a certain value. Better "relationship between the index and recent large-scale atmospheric circulation" ("past" sounds a bit like historical)

L84: Univariate bias correction might induce artefacts when studying compound events (Zscheischler et al., 2019), this might be highly relevant here. Consider applying a multivariate bias correction approach.

Figure 2: "blue lines correspond to high flows" is unclear. There is one blue line in the precipitation figure and a red line in the temperature figure. It looks as if they would just correspond to the mean of the boxplots. It would be surprised if the highflows would align so well with the precipitation amounts. Please clarify.

Section 3.2: I assume this is after bias correction?

Figure 4 and following: are these comparisons on the same spatial grid?

Figure 8: why do so few events result in high streamflow?

Consider reporting the events as relative numbers (e.g. sections 3.2, 3.3). This might be more intuitive as it is easier for the reader to put the occurrence probability into context.

Some method description appear in the results, e.g. L 215 and following.

L220: I assume TOT are the events as simulated with the hydrological model? This should be mentioned somewhere explicitly.

References: Zscheischler, J., Fischer, E. M. & Lange, S. The effect of univariate bias adjustment on multivariate hazard estimates. Earth Syst. Dyn. 10, 31–43 (2019).

---

## Author Comment (AC1) · 13 Dec 2019

**Response to Reviewer #1:**

**We would like to thank the reviewer for their productive comments on our manuscript. Please find our point by point answer as follow:**

This manuscript presents an index to describe warm, heavy rain events during winter in southern Ontario. This index is then used to evaluate the occurrence of such events with different large scale circulation patterns and projections for the future. The paper presents a thorough analysis and I appreciate the detailed discussion included in Section 4. Some concerns are listed below.

My main concern is that the authors have not provided sufficient justification for the need to create a new index. The introduction focuses on rain on snow events, but this new index does not consider snow cover. This disconnect needs to be more fully explained. Why is the proposed index better than those used in previous literature?

**The main goal of this study was to understand how the frequency of winter weather extreme events (temperature and precipitation), simulated by CRCM5-LE, is modulated by large scale atmospheric circulation. Studying such events are mostly relevant if they have societal implications. A strong shift in high flows occurrence from spring to winter was observed recently in southern Ontario and is expected to continue in the future. Therefore, we decided to define temperature and precipitation thresholds that may explain the generation of high flows in several watersheds in Ontario. Defining an index based also on snow would have been interesting but is not in the scope of this study. A major originality of the study was the calculation of future weather regimes for each member of CanESM2-LE to investigate how the variability of atmospheric circulation will impact the winter weather extremes. The weather regime of a given day impacts directly local temperature and precipitation conditions and investigating also snowmelt adds a level of complexity. Indeed, snowmelt of a given day depends also on the atmospheric conditions occurring weeks before the extreme events (major snowfalls following by cold conditions keeping the snow on the ground). Therefore, weather regimes of these days would also need to be investigated. The need of studying the sequence of weather regimes occurring prior to a high flow event in future studies was discussed at the end of section 4.4.**

**Moreover, when using snowmelt in the index (With Rain on snow index (ROS) for example) some questions are arising. ROS index does not take into consideration the rain only events while it can have a significant impact on high flows. The occurrence of ROS events is decreasing in the Great Lakes region because of an increase in days without snow on the ground (Jeong and Sushama, 2018), but this doesn't lead to a decrease in high flows. Our index takes into consideration rain events even with the absence of snow on the ground, conditions that are expecting to become more frequent in the future. The proposed index is not meant to be better than ROS but is adapted to the study of weather extreme events simulated by CRCM5-LE and how these events are impacted by large scale atmospheric circulation. As stated in the**

discussion, ROS and our index can be studied together to understand the future evolution of different hydrometeorological extreme events (Rain only, rain on snow, snowmelt).

**PRMS hydrological model was previously set up in this region (Champagne et al., 2019) which gave us the opportunity to discuss the shortcoming of this index to explain high flows events. We used PRMS to investigate how the future evolution of high flows is corelated to the future evolution of weather extreme events. But the objective was not to create an index using snow data from PRMS output. Nevertheless, to strengthen the discussion around snowmelt, we propose to add a figure showing the evolution of snowmelt between 1961-1990 and 2026-2055 corresponding to each weather pattern. This will feed the discussion on the need to study the impact of the sequences of weather regimes on snowmelt and high flows.**

Considering this is a special issue on large ensembles, I think there should be more discussion about how this work is taking advantage of the large ensemble used here.

**The main objective of this work was to assess how the internal variability of climate has an impact on the variability of local meteorological extreme events in southern Ontario. To investigate these extreme events in a small region such as southern Ontario, high resolution simulations are required. A regional large ensemble such as CRCM5-LE has the double advantage of simulating the local climate and that each member of CRCM5-LE can be related to large-scale atmospheric circulation from its corresponding CanESM2-LE member. Statistical stochastic methods to downscale GCM data could also represent well the local climate but cannot be related to a corresponding large-scale atmospheric circulation. Few sentences explaining the double advantage of a regional large ensemble will be added in the method section.**

Specific comments:

Line 52: Is the "precipitation" here referring to rain or snow or both?

**Precipitation is referring to both rain and snow. A mention ''Rain and snow'' will be added to improve the clarity of the sentence.**

Line 84: Was only the future period bias corrected?

**The bias correction was applied for the entire period (past included). The mention ''Future'' will be removed to avoid confusions.**

Line 87: Is precipitation frequency also modeled with a gamma distribution? This makes more sense for intensity.

**The method developed by Ines and Hansen (2006) involves two steps in the bias correction of precipitation. For each month, the first step consists on truncating the distribution of the modelled frequency of daily precipitation in order to match the observed distribution of precipitation frequency. The second step used the truncated distribution of precipitation**

intensity into a gamma distribution fitted to the observed intensity distribution. For clarity, these explanations on the bias correction method will be added to the manuscript.

Line 138-141: What effect would the trends within these two periods have?

**The Z500 anomalies were calculated and normalized for two distinguished period (62 years in the past and 62 years in the future) to avoid the low frequency variability (Here positive trend in Z500) to be disproportionate compare to the high frequency variability. However, this method does not remove all the low frequency Z500 trends. Following this comment, we investigated the change in occurrence of regimes within these two periods and results show a large increase in occurrence of regime HP within the 2026-2055 period. Therefore, the regimes HP are occurring mostly at a period when the conditions are warmer (At the end of the 2026-2055 period), which overestimates HP average temperature and number of warm events. To avoid this artefact, we decided to slightly change our original method and normalize the anomalies in period of 30 years before calculating the regimes. This method minimizes the impact of low frequency on each regime average climate while keeping sufficient periods length (30 years) for the calculations of the anomalies. The number of warm events corresponding to regimes HP in the future period is clearly decreased with this new method. The results and discussion will be modified according to the new figures made with the new method. We will also modify the figure 9 *(DJF change in number of precipitation and warm events between the historical (1961-1990) and the future period (2026-2055) for the 50 members CRCM5-LE average)* and turned it into change in number of extreme events per occurrence of weather regimes. Indeed we want to remove the impact of the 50-members average change in occurrence of weather regimes on extreme events and study only the variability between members. The goal of this figure is to rather investigate the stability between the past and the future regarding the number of weather extreme events occurring for each weather regime. The impact of change in occurrence of weather regimes on weather extreme events is anyway investigated through the variability between the members of CanESM2-LE/CRCM5-LE (Figure 11 and 12, Table 1 and 2).**

Section2.4/Section 3.1: Were the top regimes determined separately for the model and reanalysis? If so, then I think the good agreement warrants more commentary. If not, can you describe more clearly how they are related?

**A principal component analysis (PCA) was first applied to the daily 20thCR Z500 anomalies calculated between 1956 and 2012 for each grid of the domain. The principal components explaining 80% of the spatial variance were identified and their eigenvectors were used to transform the original time series into the principal components time series. This daily principal components time series was then classified using the k-means algorithm. The k-means algorithm identifies iteratively classes centroids that maximize the interclass variance and minimize the intraclass variance. To classify the daily Z500 for each member of CanESM2, the same regimes identified with 20thCR were used. First, the same eigenvectors identified in the PCA using 20thCR Z500 were used to calculate the principal components time series for each**

**member of CanESM2. Then, the k-means classes centroids identified with 20thCR were used to classify the principal components time series for each member of CanESM2 large ensemble. This more accurate description of the CanESM2 regimes identifications will be added to the manuscript.**

Section 3.2: Is the modelled and bias-corrected data being compared against the dataset used for the bias correction target? If so, can you add a brief discussion on the implications?

**The modelled and bias corrected data have not been explicitly compared in the study. The extreme events identified from the raw data from CRCM5-LE will be compared to the extremes calculated with the observations. A new figure (Figure R1) will be include in supplementary materials to show the need in applying bias correction. This figure show that the difference between simulations and observations is much higher when using the raw data (Figure R1) compared to the bias corrected data (Figure 4 main manuscript).**

Section 3.3: What does the model bias in the South regime mean for these projections?

**The model bias between simulated Z500 and observed Z500 in the regime South in the past is likely to remain similar in the future because the same simulated dataset (CanESM2-LE) is used for the future period. The number of events for a regime South calculated with CanESM2 will likely be overestimated compare to a pattern that would look like the 20thCR regime South. However what is relevant in this study is to understand why the atmospheric circulation of the regime South using CanESM2 anomalies produces weather extreme events and how the change in number of regime South occurrence will modulate the number of extreme events. We will add a new column to the figure 3 showing the future evolution of Z500 corresponding to each regime. A discussion on the impact of a change in weather regimes Z500 anomalies will be added to the discussion.**

How robust are the streamflow projections when they do not account for changes in snowfall/snowpack or associated feedbacks?

**A new figure showing the evolution of snowmelt corresponding to each extreme weather events will be added to the manuscript. This figure will provide more discussion on the difference between weather extreme events and high flows.**

Line 229: Do you mean the magnitude of the correlations here?

**We meant the magnitude of the correlations. This will be added in the new version of the manuscript.**

Line 237: Can you clarify what is meant by the combination of weather regimes? The value of using these combinations instead of just changes in individual regimes is not clear.

**A combination of two weather regimes has been done by summing the seasonal occurrence of these two weather regimes. If for a given winter the regime HP occurs 20 times and the regime South 15 times, there combination will be 35. The goal of using these combinations is to**

**identify the impact of a combination of weather patterns occurring the same winter on weather extremes and high-flows. The weather patterns are linked to each other because are a discretization of a continuous process (Atmospheric circulation). As stated in the discussion, a given pattern recurrently succeed to the same patterns because the systems (cyclones and anticyclones) are following a general direction (West to east). Therefore, the weather regimes are not independent from each other. A combination of weather regimes occurrence shows the impact of a simulteanously large occurrence of two weather regimes on hydrometeorological extremes. It is particularly valuable to understand their impact on high-flows because atmospheric conditions days before the events are also relevant to explain the generation of high flows events (i.e. Formation of Snowpack). A clearer explanation on what is a combination of weather regimes and its purpose will be added to the new version of the manuscript.**

Figure 1: I found it very difficult to orient myself in this figure. It is hard to discern the coastlines of the Great Lakes from the contours in the large scale figure and the legend covers the land separating two lakes. It would be very helpful to label the lakes and key places here, especially since so much of the later discussion is very specific about local geography.

**This figure will be replaced by a new figure with a modified large-scale figure, a clearer separation between Lake Erie and Ontario and the name of the Lakes (Figure R2).**

Figure 2: Why is the scale of the horizontal axis nonlinear? With the current spacing and connecting line, this time series is misleading.

**The scale of horizontal axis is not linear because represent different high flow events. The connecting lines representing the average will be removed in the new figure as they are confusing and do not give any valuable information (Figure R3).**

Figures 4-7: The figure label ("simulations minus observations") does not agree with the caption ("CanGRD minus CRCM5-LE")

**The words simulations and observations will be added to the legend.**

Figure 8: Are the ensemble values pooled or averaged?

**The ensemble values are pooled. This information will be added to the manuscript.**

Line 462: This paper should be cited as Jeong and Sushama

**This modification will be done**

References:

Champagne, O., Arain, M. A. and Coulibaly, P.: Atmospheric circulation amplifies shift of winter streamflow in Southern Ontario, Journal of Hydrology, 124051, doi:10.1016/j.jhydrol.2019.124051, 2019.

Ines, A. V. M. and Hansen, J. W.: Bias correction of daily GCM rainfall for crop simulation studies, Agricultural and Forest Meteorology, 138(1–4), 44–53, doi:10.1016/j.agrformet.2006.03.009, 2006.

Jeong, D. and Sushama, L.: Rain-on-snow events over North America based on two Canadian regional climate models, Climate Dynamics, 50(1–2), 303–316, doi:10.1007/s00382-017-3609-x, 2018.

---

## Author Comment (AC2) · 13 Dec 2019

**Response Reviewer #2**

The authors investigate the occurrence of rain on snow events with a single model ensemble. Overall this is a very interesting paper with a good application of the bottomup approach that has recently been endorsed for studying compound events.

Using the compound exceedance of a precipitation and temperature threshold seems relatively naive, given that large increases in temperature in the future will lead to very different snow cover patterns, a key determinant of ROS events. A more appropriate variable than temperature seems to me the difference in surface snow amount between consecutive days, which could be used as a proxy for snowmelt. This is available from the model output. With this it should be possible to build a better compound index that should also be more reliable in future projections.

**The main goal of this study was to understand how the frequency of winter weather extreme events (temperature and precipitation), simulated by CRCM5-LE, is modulated by large scale atmospheric circulation. Studying such events are mostly relevant if they have societal implications. A strong shift in high flows occurrence from spring to winter was observed recently in southern Ontario and is expected to continue in the future. Therefore, we decided to define temperature and precipitation thresholds that may explain the generation of high flows in several watersheds in Ontario. Defining an index based also on snow would have been interesting but is not in the scope of this study. A major originality of the study was the calculation of future weather regimes for each member of CanESM2-LE to investigate how the variability of atmospheric circulation will impact the winter weather extremes. The weather regime of a given day impacts directly local temperature and precipitation conditions and investigating also snowmelt adds a level of complexity. Indeed, snowmelt of a given day depends also on the atmospheric conditions occurring weeks before the extreme events (major snowfalls following by cold conditions keeping the snow on the ground). Therefore, weather regimes of these days would also need to be investigated. The need of studying the sequence of weather regimes occurring prior to a high flow event in future studies was discussed at the end of section 4.4.**

**Moreover, when using snowmelt in the index (With Rain on snow index (ROS) for example) some questions are arising. ROS index does not take into consideration the rain only events while it can have a significant impact on high flows. The occurrence of ROS events is decreasing in the Great Lakes region because of an increase in days without snow on the ground** (Jeong and Sushama, 2018)**, but this doesn't lead to a decrease in high flows. Our index takes into consideration rain events even with the absence of snow on the ground, conditions that are expecting to become more frequent in the future. The proposed index is not meant to be better than ROS but is adapted to the study of weather extreme events simulated by CRCM5-LE and how they are impacted by large scale atmospheric circulation. As stated in the discussion, ROS and our index can be studied together to understand the future evolution of different hydrometeorological extreme events (Rain only, rain on snow, snowmelt).**

**PRMS hydrological model was previously set up in this region (Champagne et al., 2019) which gave us the opportunity to discuss the shortcoming of this index to explain high flows events.. We used PRMS to investigate how the future evolution of high flows is corelated to the future evolution of weather extreme events. But the objective was not to create an index using snow data from PRMS output. Nevertheless, to strengthen the discussion around snowmelt, we propose to add a figure showing the evolution of snowmelt between 1961-1990 and 2026-2055 corresponding to each weather pattern. This will feed the discussion on the need to study the impact of the sequences of weather regimes on snowmelt and high flows.**

I suspect streamflow is very non-gaussian distributed. In particular, it's asymmetric and bounded from below. Taking the mean +3 standard deviations as an indicator for extremes is thus very unintuitive and not really appropriate for such a distribution. I would suggest to use a high percentile (e.g. above the 99th percentile, or something similar, could also be more extreme). This can then also be translated easily into a return period.

**The mean +3 standard deviations will be changed to 99$^{th}$ percentile in the identification of high flows for each watershed.**

Would it be an option to us only the weather patterns based on the observations and classify the models according to those? This might reduce differences between models and observations with respect to the occurrence rate of heavy precip and warm events (the authors discuss this point in sec 4.1).

**The models were classified according to the weather patterns calculated with the observations (20thCR reanalyses). The daily Z500 anomalies from the observations were first transformed by principal component analysis (PCA) keeping 80% of the spatial variance. The principal components identified were then classified into recurrent weather patterns using a k-means algorithm. The eigenvectors of the PCA as well as the k-means centroids of the patterns identified using the observations, are used to identify the weather regimes for each member of CanESM2-LE. The explanations of the method used to calculate the CanESM2 weather regimes will be improved in the new manuscript**

Please mention somewhere explicitly how the compound index is defined. Is it just the occurrence of events where both temperature and precipitation exceed a certain threshold? Or the number of such occurrences?

**The compound index is simply defined by the number of days with a temperature exceeding 5 degrees and precipitation exceeding 10mm. The information will be explicitly added to the method section.**

Minor comments: I would recommend the authors to do a thorough spell check and grammar check. There are a number of minor grammatical errors and typos in the text.

**A spell and grammar check will be done for the entire manuscript.**

L 49: start new paragraph

L59: "preconized" ?

L67: "contributes to": maybe better: "explains the variability of"

L69: "occurrence of the index": an index does not occur, it has a certain value. Better "relationship between the index and recent large-scale atmospheric circulation" ("past" sounds a bit like historical)

**These modifications will be done as suggested**

L84: Univariate bias correction might induce artefacts when studying compound events (Zscheischler et al., 2019), this might be highly relevant here. Consider applying a multivariate bias correction approach.

**The bias correction approach used in this study was used in a previous study in the area (Champagne et al., 2019). For consistency with this previous study, the same bias correction technique was applied. We also identified the number of extreme events using the raw data (Figure R1) and found a higher difference between simulations and observation compared to the bias corrected data (Figure 4 in the main manuscript). These results are showing that this bias correction method is satisfactory. A reference to a multivariate bias correction approach will be added to the discussion.**

Figure 2: "blue lines correspond to high flows" is unclear. There is one blue line in the precipitation figure and a red line in the temperature figure. It looks as if they would just correspond to the mean of the boxplots. It would be surprised if the highflows would align so well with the precipitation amounts. Please clarify.

**These blue and red lines correspond to the mean of the boxplots. These lines are not giving valuable information and will be removed for clarity.**

Section 3.2: I assume this is after bias correction?

**Yes the results are given using bias correction data. This information will be added to the manuscript**

Figure 4 and following: are these comparisons on the same spatial grid?

**These comparisons are on the same spatial grid because the bias correction was performed at each observed grid point. The modelled grid-point the closest from each observed grid point was identified and the corresponding temperature and precipitation were bias corrected. These bias corrected data are represented at each observed grid point in the figures.**

Figure 8: why do so few events result in high streamflow?

**Few events result in high flows because even though the index is a condition to produce a high flow event the generation of high flows also needs other conditions (other rain events in the previous days, snowmelt amount). This will be more discussed in the manuscript.**

Consider reporting the events as relative numbers (e.g. sections 3.2, 3.3). This might be more intuitive as it is easier for the reader to put the occurrence probability into context.

**The relative numbers will be added to the manuscript.**

Some method description appear in the results, e.g. L 215 and following.

**These elements of methods will be added to the method section.**

L220: I assume TOT are the events as simulated with the hydrological model? This should be mentioned somewhere explicitly.

**The mention ''simulated by PRMS'' will be added to the manuscript**

**References:**

Champagne, O., Arain, M. A. and Coulibaly, P.: Atmospheric circulation amplifies shift of winter streamflow in Southern Ontario, Journal of Hydrology, 124051, doi:10.1016/j.jhydrol.2019.124051, 2019.

Jeong, D. and Sushama, L.: Rain-on-snow events over North America based on two Canadian regional climate models, Climate Dynamics, 50(1–2), 303–316, doi:10.1007/s00382-017-3609-x, 2018.

Zscheischler, J., Fischer, E. M. and Lange, S.: The effect of univariate bias adjustment on multivariate hazard estimates, Earth System Dynamics, 10(1), 31–43, doi:10.5194/esd-10-31-2019, 2019.

---

## Author Response (AR1)

[revised manuscript text omitted]

**We would like to thank again the reviewers for their constructive comments. Please find**
**our point by point answer as follow:**
**Response to Reviewer #1:**
This manuscript presents an index to describe warm, heavy rain events during winter in
southern Ontario. This index is then used to evaluate the occurrence of such events with
different large scale circulation patterns and projections for the future. The paper presents a
thorough analysis and I appreciate the detailed discussion included in Section 4. Some concerns
are listed below.
My main concern is that the authors have not provided sufficient justification for the need to
create a new index. The introduction focuses on rain on snow events, but this new index does
not consider snow cover. This disconnect needs to be more fully explained. Why is the
proposed index better than those used in previous literature?
**The main goal of this study was to understand how the frequency of winter weather**
**extreme events (temperature and precipitation), simulated by CRCM5-LE, is modulated**
**by large scale atmospheric circulation. Studying such events are mostly relevant if they**
**have societal implications. A strong shift in high flows occurrence from spring to winter**
**was observed recently in southern Ontario and is expected to continue in the future.**
**Therefore, we decided to define temperature and precipitation thresholds that may**
**explain the generation of high flows in several watersheds in Ontario. Defining an index**
**based also on snow would have been interesting but is not in the scope of this study. A**
**major originality of the study was the calculation of future weather regimes for each**
**member of CanESM2-LE to investigate how the variability of atmospheric circulation**
**will impact the winter weather extremes. The weather pattern of a given day impacts**
**directly local temperature and precipitation conditions and investigating also snowmelt**
**adds a level of complexity. Indeed, snowmelt of a given day depends also on the**
**atmospheric conditions occurring weeks before the extreme events (major snowfalls**
**following by cold conditions keeping the snow on the ground). Therefore, weather**
**regimes of these days would also need to be investigated. The need of studying the**
**sequence of weather regimes occurring prior to a high flow event in future studies was**
**discussed at the end of section 4.3.**
**Moreover, when using snowmelt in the index (With Rain on snow index (ROS) for**
**example) some questions are arising. ROS index does not take into consideration the rain**
**only events while it can have a significant impact on high flows. The occurrence of ROS**

**events is decreasing in the Great Lakes region because of an increase in days without snow on the ground (Jeong and Sushama, 2018), but this doesn't lead to a decrease in high flows. Our index takes into consideration rain events even with the absence of snow on the ground, conditions that are expecting to become more frequent in the future. The proposed index is not meant to be better than ROS but is adapted to the study of weather extreme events simulated by CRCM5-LE and how these events are impacted by large scale atmospheric circulation. As stated in the section 4.4, ROS and our index can be studied together to understand the future evolution of different hydrometeorological extreme events (Rain only, rain on snow, snowmelt).**

**PRMS hydrological model was previously set up in this region** (Champagne et al., 2019a) **which gave us the opportunity to discuss the ability of this index to explain high flows events. We used PRMS to investigate how the future evolution of high flows is corelated to the future evolution of weather extreme events. But the objective was not to create an index using snow data from PRMS output. Nevertheless, to strengthen the discussion around snowmelt, we added a figure showing the evolution of snowmelt between 1961-1990 and 2026-2055 corresponding to each weather pattern (Figure 11 of the new manuscript).**

Considering this is a special issue on large ensembles, I think there should be more discussion about how this work is taking advantage of the large ensemble used here.

**The main objective of this work was to assess how the internal variability of climate has an impact on the variability of local meteorological extreme events in southern Ontario. To investigate these extreme events in a small region such as southern Ontario, high resolution simulations are required. A regional large ensemble such as CRCM5-LE has the double advantage of simulating the local climate and that each member of CRCM5-LE can be related to large-scale atmospheric circulation from its corresponding CanESM2-LE member. Statistical stochastic methods to downscale GCM data could also represent well the local climate but cannot be related to a corresponding large-scale atmospheric circulation. Few sentences explaining the double advantage of a regional large ensemble was added in the section 2.1.**

Specific comments:

Line 52: Is the "precipitation" here referring to rain or snow or both?

**Precipitation is referring to both rain and snow. A mention ''Rain and snow'' was added to improve the clarity of the sentence.**

Line 84: Was only the future period bias corrected?

**The bias correction was applied for the entire period (past included). The mention ''Future'' has been removed to avoid confusions.**

Line 87: Is precipitation frequency also modeled with a gamma distribution? This makes more sense for intensity.

**The method developed by Ines and Hansen (2006) involves two steps in the bias correction of precipitation. For each month, the first step consists on truncating the distribution of the modelled frequency of daily precipitation in order to match the observed distribution of precipitation frequency. The second step used the truncated distribution of precipitation intensity into a gamma distribution fitted to the observed intensity distribution. For clarity, these explanations on the bias correction method have been added to the manuscript.**

Line 138-141: What effect would the trends within these two periods have?

**The Z500 anomalies were calculated and normalized for two distinguished period (55 years in the past and 55 years in the future) to avoid the low frequency variability (Here positive trend in Z500) to be disproportionate compare to the high frequency variability. However, this method does not remove all the low frequency Z500 trends. Following this comment, we investigated the change in occurrence of regimes within these two periods and results show a large increase in occurrence of regime HP within the 2026-2055 period. Therefore, the regimes HP are occurring mostly at a period when the conditions are warmer (At the end of the 2026-2055 period), which overestimates HP average temperature and number of warm events. To avoid this artefact, we decided to slightly change our original method and normalize the anomalies in period of 30 years before calculating the regimes. This method minimizes the impact of low frequency on average climate of each regime while keeping sufficient periods length (30 years) for the calculations of the anomalies. The results were not significantly different with this new method and therefore the discussion remains very similar to the first version of the manuscript.**
**We also modified the figure 9** *(DJF change in number of precipitation and warm events between the historical (1961-1990) and the future period (2026-2055) for the 50 members*

*CRCM5-LE average)* **and turned it into change in number of extreme events per occurrence of weather regimes (Figure 8 in the new manuscript). The objective is to remove the impact of the 50-members average change in occurrence of weather regimes on extreme events and study only the variability between members. The goal of this figure is to investigate the stability between the past and the future regarding the number of weather extreme events occurring for each weather regime. The impact of change in occurrence of weather regimes on weather extreme events were anyway investigated in the section 3.4 (Figure 11 and 12, Table 1 and 2).**

Section2.4/Section 3.1: Were the top regimes determined separately for the model and reanalysis? If so, then I think the good agreement warrants more commentary. If not, can you describe more clearly how they are related?

**A principal component analysis (PCA) was first applied to the daily 20thCR Z500 anomalies calculated between 1956 and 2012 for each grid of the domain. The principal components explaining 80% of the spatial variance were identified and their eigenvectors were used to transform the original time series into the principal components time series. This daily principal components time series was then classified using the k-means algorithm. The k-means algorithm identifies iteratively classes centroids that maximize the interclass variance and minimize the intraclass variance. To classify the daily Z500 for each member of CanESM2, the same regimes identified with 20thCR were used. First, the same eigenvectors identified in the PCA using 20thCR Z500 were used to calculate the principal components time series for each member of CanESM2. Then, the k-means classes centroids identified with 20thCR were used to classify the principal components time series for each member of CanESM2 large ensemble. For a better understanding of the method, a more accurate description of the CanESM2 regimes identifications were added to the section 2.3.**

Section 3.2: Is the modelled and bias-corrected data being compared against the dataset used for the bias correction target? If so, can you add a brief discussion on the implications?

**The modelled and bias corrected data have not been explicitly compared in the first version of the manuscript. Following this comment, the extreme events identified from the raw data from CRCM5-LE have been compared to the extremes calculated with the observations and a figure has been included in supplementary materials (Figure S2). This figure show that the difference between simulations and observations is much higher when using the raw data (Figure S2) compared to the bias corrected data (Figure S1 in supplementary materials).**

Section 3.3: What does the model bias in the South regime mean for these projections?

**The model bias between simulated Z500 and observed Z500 in the regime South in the past is likely to remain similar in the future because the same simulated dataset (CanESM2-LE) is used for the future period. The number of events for a regime South calculated with CanESM2 will likely be overestimated compare to a pattern that would look like the 20thCR regime South. However what is relevant in this study is to understand why the atmospheric circulation of the regime South using CanESM2 anomalies produces weather extreme events and how the change in number of regime South occurrence will modulate the number of extreme events. We added a new column to the figure 3 showing the future evolution of Z500 corresponding to each regime.**

How robust are the streamflow projections when they do not account for changes in snowfall/snowpack or associated feedbacks?

**The streamflow projections were computed by the model PRMS. PRMS accounts for change in snowfall and snowpack to calculate streamflow.**

Line 229: Do you mean the magnitude of the correlations here?

**We meant the magnitude of the correlations. This was added to the new version of the manuscript.**

Line 237: Can you clarify what is meant by the combination of weather regimes? The value of using these combinations instead of just changes in individual regimes is not clear.

**A combination of two weather regimes has been done by summing the seasonal occurrence of these two weather regimes. If for a given winter the regime HP occurs 20 times and the regime South 15 times, there combination will be 35. The goal of using these combinations is to identify the impact of a combination of weather patterns occurring the same winter on weather extremes and high-flows. The weather patterns are linked to each other because are a discretization of a continuous process (Atmospheric circulation). As stated in the discussion, a given pattern recurrently succeed to the same patterns because the systems (cyclones and anticyclones) are following a general direction (West to east). Therefore, the weather regimes are not independent from each other. A combination of weather regimes occurrence shows the impact of a simultaneously large occurrence of two weather regimes on hydrometeorological extremes. It is particularly valuable to understand their impact on high-flows because atmospheric conditions days before the**

**events are also relevant to explain the generation of high flows events (i.e. Formation of Snowpack). A clearer explanation on what is a combination of weather regimes and its purpose was added to the section 3.4.**

Figure 1: I found it very difficult to orient myself in this figure. It is hard to discern the coastlines of the Great Lakes from the contours in the large scale figure and the legend covers the land separating two lakes. It would be very helpful to label the lakes and key places here, especially since so much of the later discussion is very specific about local geography.

**This figure was replaced by a new figure with a modified large-scale figure, a clearer separation between Lake Erie and Ontario and the name of the Lakes.**

Figure 2: Why is the scale of the horizontal axis nonlinear? With the current spacing and connecting line, this time series is misleading.

**The scale of horizontal axis is not linear because represent different high flow events. The connecting lines representing the average have been removed in the new figure as they are confusing and do not give any valuable information.**

Figures 4-7: The figure label ("simulations minus observations") does not agree with the caption ("CanGRD minus CRCM5-LE")

**The words simulations and observations have been added to the legend.**

Figure 8: Are the ensemble values pooled or averaged?

**The ensemble values are pooled. This information has been added to the manuscript.**

Line 462: This paper should be cited as Jeong and Susham

**This modification has been done**

The authors investigate the occurrence of rain on snow events with a single model ensemble. Overall this is a very interesting paper with a good application of the bottomup approach that has recently been endorsed for studying compound events.

Using the compound exceedance of a precipitation and temperature threshold seems relatively naive, given that large increases in temperature in the future will lead to very different snow cover patterns, a key determinant of ROS events. A more appropriate variable than temperature seems to me the difference in surface snow amount between consecutive days, which could be used as a proxy for snowmelt. This is available from the model output. With this it should be possible to build a better compound index that should also be more reliable in future projections.

**The main goal of this study was to understand how the frequency of winter weather extreme events (temperature and precipitation), simulated by CRCM5-LE, is modulated by large scale atmospheric circulation. Studying such events are mostly relevant if they have societal implications. A strong shift in high flows occurrence from spring to winter was observed recently in southern Ontario and is expected to continue in the future. Therefore, we decided to define temperature and precipitation thresholds that may explain the generation of high flows in several watersheds in Ontario. Defining an index based also on snow would have been interesting but is not in the scope of this study. A major originality of the study was the calculation of future weather regimes for each member of CanESM2-LE to investigate how the variability of atmospheric circulation will impact the winter weather extremes. The weather pattern of a given day impacts directly local temperature and precipitation conditions and investigating also snowmelt adds a level of complexity. Indeed, snowmelt of a given day depends also on the atmospheric conditions occurring weeks before the extreme events (major snowfalls following by cold conditions keeping the snow on the ground). Therefore, weather regimes of these days would also need to be investigated. The need of studying the sequence of weather regimes occurring prior to a high flow event in future studies was discussed at the end of section 4.4.**

**Moreover, when using snowmelt in the index (With Rain on snow index (ROS) for example) some questions are arising. ROS index does not take into consideration the rain only events while it can have a significant impact on high flows. The occurrence of ROS events is decreasing in the Great Lakes region because of an increase in days without snow on the ground (Jeong and Sushama, 2018), but this doesn't lead to a decrease in**

**high flows. Our index takes into consideration rain events even with the absence of snow on the ground, conditions that are expecting to become more frequent in the future. The proposed index is not meant to be better than ROS but is adapted to the study of weather extreme events simulated by CRCM5-LE and how these events are impacted by large scale atmospheric circulation. As stated in the section 4.5, ROS and our index can be studied together to understand the future evolution of different hydrometeorological extreme events (Rain only, rain on snow, snowmelt).**

**PRMS hydrological model was previously set up in this region** (Champagne et al., 2019a) **which gave us the opportunity to discuss the ability of this index to explain high flows events. We used PRMS to investigate how the future evolution of high flows is corelated to the future evolution of weather extreme events. But the objective was not to create an index using snow data from PRMS output. Nevertheless, to strengthen the discussion around snowmelt, we added a figure showing the evolution of snowmelt between 1961-1990 and 2026-2055 corresponding to each weather pattern (Figure 11 of the new manuscript).**

I suspect streamflow is very non-gaussian distributed. In particular, it's asymmetric and bounded from below. Taking the mean +3 standard deviations as an indicator for extremes is thus very unintuitive and not really appropriate for such a distribution. I would suggest to use a high percentile (e.g. above the 99th percentile, or something similar, could also be more extreme). This can then also be translated easily into a return period.

**The mean +3 standard deviations was changed to 99$^{th}$ percentile in the entire manuscript.**

Would it be an option to us only the weather patterns based on the observations and classify the models according to those? This might reduce differences between models and observations with respect to the occurrence rate of heavy precip and warm events (the authors discuss this point in sec 4.1).

**The models were classified according to the weather patterns calculated with the observations (20thCR reanalyses). The daily Z500 anomalies from the observations were first transformed by principal component analysis (PCA) keeping 80% of the spatial variance. The principal components identified were then classified into recurrent weather patterns using a k-means algorithm. The eigenvectors of the PCA as well as the k-means centroids of the patterns identified using the observations, are used to identify the weather regimes for each member of CanESM2-LE. The explanations of the method used to calculate the CanESM2 weather regimes was improved in the section 2.3.**

Please mention somewhere explicitly how the compound index is defined. Is it just the occurrence of events where both temperature and precipitation exceed a certain threshold? Or the number of such occurrences?

**The compound index is simply defined by the number of days with a temperature exceeding 5 degrees and precipitation exceeding 10mm. The information was explicitly added to the method section (Section 2.2).**

Minor comments: I would recommend the authors to do a thorough spell check and grammar check. There are a number of minor grammatical errors and typos in the text.

**A spell and grammar check will be done for the entire manuscript.**

L 49: start new paragraph

L59: "preconized" ?

L67: "contributes to": maybe better: "explains the variability of"

L69: "occurrence of the index": an index does not occur, it has a certain value. Better "relationship between the index and recent large-scale atmospheric circulation" ("past" sounds a bit like historical)

**These modifications were done as suggested**

L84: Univariate bias correction might induce artefacts when studying compound events (Zscheischler et al., 2019), this might be highly relevant here. Consider applying a multivariate bias correction approach.

**The bias correction approach used in this study was used in a previous study in the area (Champagne et al., 2019a). For consistency with this previous study, the same bias correction technique was applied. We also identified the number of extreme events using the raw data (Supplementary materials Figure S2) and found a higher difference between simulations and observation compared to the bias corrected data (Supplementary materials Figure S1). These results are showing that this bias correction method is satisfactory. A reference to a multivariate bias correction approach was added to the discussion (Section 4.1)**

**Please note that the figure S1 was previously in the manuscript (Figure 4 in the first**
**manuscript). This figure was moved to supplementary material for an easy visual**
**comparison between the events calculated from bias corrected data and from raw data.**
**The plots in figure S1 are still in the manuscript and have been added to figure 4, 5 and**
**6 (Column ''all'').**
Figure 2: "blue lines correspond to high flows" is unclear. There is one blue line in the
precipitation figure and a red line in the temperature figure. It looks as if they would just
correspond to the mean of the boxplots. It would be surprised if the highflows would align so
well with the precipitation amounts. Please clarify.
**These blue and red lines correspond to the mean of the boxplots. These lines are not giving**
**valuable information and were removed for clarity.**
Section 3.2: I assume this is after bias correction?
**Yes the results are given using bias correction data. This information has been added to**
**the manuscript**
Figure 4 and following: are these comparisons on the same spatial grid?
**These comparisons are on the same spatial grid because the bias correction was**
**performed at each observed grid point. The modelled grid-point the closest from each**
**observed grid point was identified and the corresponding temperature and precipitation**
**were bias corrected. These bias corrected data are represented at each observed grid**
**point in the figures.**
Figure 8: why do so few events result in high streamflow?
**Few events result in high flows because even though the index is a condition to produce a**
**high flow event the generation of high flows also needs other conditions (other rain events**
**in the previous days, snowmelt amount). This discussion has been added to the**
**manuscript (Section 4.5).**
Consider reporting the events as relative numbers (e.g. sections 3.2, 3.3). This might be more
intuitive as it is easier for the reader to put the occurrence probability into context.
**The relative numbers have been added to the manuscript.**

Some method description appear in the results, e.g. L 215 and following.

**These elements of methods were put in the method section.**

L220: I assume TOT are the events as simulated with the hydrological model? This should be mentioned somewhere explicitly.

**The mention ''simulated by PRMS'' was added to the manuscript**

---

## Author Response (AR2)

**We would like to thank the editor for accepting our paper and for giving us constructive comments. Please find our answers followed by the annotated manuscript:**

Dear authors,

I am happy to accept your manuscript for publication subject to the small amount of revisions outlined below.

Kind regards,
Nicola Maher

Note the lines are based on the document esd-2019-56-author_response-version1
Line 9 terms
Line 11 , but
value of the
slightly increases

**These changes have been done in the manuscript**

332-333 these lines are really confusing
onwards - the use of change is confusing perhaps 'the change ' would clarify this paragraph

**This sentence was reformulated to increase clarity**

can a correlation improve, this should be increase

**"improve" was changed to "increase"**

Same paragraph it should be correlated with not to

**Correlated to was changed to correlated with in the entire manuscript**

, but
change in back to of

**These changes have been done**

423 use of word apprehended is odd

**"apprehended" was replaced by "driven"**

425 , but
427 the occurrence
438 that produce
438 increases
440, but
454 could be
468 to so focus on

**Changed have done as suggested**

472 are you saying your own method is questionable?

**Following reviewers comments it seemed important to highlight the method limits. For**
**clarity, "The relevance of" was then replaced by "the method".**

[revised manuscript text omitted]